# The Prevalence of High- and Low-Risk Types of HPV in Patients with Squamous Cell Carcinoma of the Head and Neck, Patients with Chronic Tonsillitis, and Healthy Individuals Living in Poland

**DOI:** 10.3390/diagnostics11122180

**Published:** 2021-11-24

**Authors:** Joanna Katarzyna Strzelczyk, Krzysztof Biernacki, Jadwiga Gaździcka, Elżbieta Chełmecka, Katarzyna Miśkiewicz-Orczyk, Natalia Zięba, Janusz Strzelczyk, Maciej Misiołek

**Affiliations:** 1Department of Medical and Molecular Biology, Faculty of Medical Sciences in Zabrze, Medical University of Silesia, 19 Jordana Str., 41-808 Zabrze, Poland; kbiernacki@sum.edu.pl (K.B.); jgazdzicka@sum.edu.pl (J.G.); 2Department of Statistics, Department of Instrumental Analysis, Faculty of Pharmaceutical Sciences in Sosnowiec, Medical University of Silesia, 30 Ostrogórska Str., 40-055 Katowice, Poland; echelmecka@sum.edu.pl; 3Department of Otorhinolaryngology and Oncological Laryngology, Faculty of Medical Sciences in Zabrze, Medical University of Silesia in Katowice, 10 C Skłodowskiej Str., 41-800 Zabrze, Poland; kmiskiewicz-orczyk@sum.edu.pl (K.M.-O.); nataliazieba@sum.edu.pl (N.Z.); maciej.misiolek@sum.edu.pl (M.M.); 4Department of Endocrinology and Neuroendocrine Tumors, Department of Pathophysiology and Endocrinology, Faculty of Medical Sciences in Zabrze, Medical University of Silesia in Katowice, 35 Ceglana Str., 40-514 Katowice, Poland; janusz.strzelczyk@sum.edu.pl

**Keywords:** human papillomavirus, head and neck squamous cell carcinoma, chronic tonsillitis, high-risk types, low-risk types

## Abstract

Human papillomavirus (HPV) is a virus with the potential to infect human epithelial cells and an etiological agent of many types of cancer, including head and neck cancer. The aim of the study was to determine the prevalence of HPV infection in patients with head and neck squamous cell carcinoma (HNSCC), patients with chronic tonsillitis, and healthy individuals, and to establish high- and low-risk HPV genotypes in these groups. The objectives also comprised the delineation of the relationship between the infection with high- or low-risk HPV subtypes and clinicopathological and demographic characteristics of the study groups. This study was composed of 76 patients diagnosed with HNSCC, 71 patients with chronic tonsillitis, and 168 cases without either of these conditions (the control group). HPV detection and identification of subtypes were performed on isolated DNA using a test which allowed detection of 33 common high-risk and low-risk HPV subtypes. The prevalence of HPV infection was 42.1%, 25.4%, and 37.5% in HNSCC, chronic tonsillitis, and control groups, respectively. HPV 16 was the most prevalent genotype in all groups and the non-oncogenic HPV 43/44 was frequent in HNSCC patients. This analysis provides insight into the prevalence of oral oncogenic and non-oncogenic HPVs in patients with head and neck cancer, patients with chronic tonsillitis and healthy individuals, and leads to the conclusion that further investigations are warranted to examine a larger cohort of patients focusing on high- and low-risk HPV genotypes. Efforts should be focused on screening and prevention strategies, and therefore, it is important to introduce tools for effective detection of HPV genotypes. Furthermore, given the role of vaccines against oral HPV infection, our observations lead to the suggestion that HPV vaccination should be of considerable importance in public health strategies.

## 1. Introduction

Human papillomavirus (HPV) is a double-stranded DNA virus with the potential to infect human epithelial cells, an etiological agent of many types of cancer [1], and the primary cause of cervical cancer [2]. In addition, HPV is a risk factor for pathogenesis of other anogenital malignancies and non-melanoma skin cancers [3,4] and is also implicated in head and neck cancer [5,6,7,8]. Incidence rates for oral and oropharyngeal cancer related to HPV infections are on the increase in some countries, especially among men [9,10,11], but the global prevalence and the genotype of HPV infection in head and neck squamous cell carcinoma (HNSCC) remain obscure [8,9]. About 200 types of HPV have been identified and classified into two major groups (i.e., high-risk and low-risk subtypes), depending on their ability to trigger neoplastic transformation [1,3], 25 types of which (1, 2, 3, 4, 6, 7, 10, 11, 13, 16, 18, 31, 32, 33, 35, 40, 45, 52, 55, 57, 58, 59, 69, 72, 73) are associated with oral lesions [12].

Despite advances in surgical treatments and chemo- and radiotherapy, there are depressing observations regarding morbidity and mortality in head and neck cancer patients. Patients with oral cancer often present with symptoms at a late stage and show high recurrence rates after treatment [13]. A rising incidence of oropharyngeal cancers was observed in developed countries, particularly at younger ages, and a significant increase in women was noted in European countries, including Poland [10]. Next to the consumption of alcohol and tobacco, HPV infection is also a risk factor for head and neck cancer [7,14]. HPV-positive HNSCC cases have lower risks of progression and are associated with better survival compared to HPV-negative cases [15]. Of note, patients with HPV-associated tumors have better clinical outcomes than patients with other risk factors [16]. To improve the identification and the prognosis of head and neck cancer, a new system of TNM classification has been introduced for HPV-positive patients [17].

Chronic tonsillitis may be defined as a sore throat lasting for 3 months and accompanied by tonsillar inflammation [18,19]. The pathogenesis is complex and may result from a persistent infection or recurrent attacks of acute tonsillitis. The common sites for chronic inflammation are the palatine and/or pharyngeal tonsils [20]. Chronic tonsillitis can lead to tonsil stones and a significant decrease in the quality of life [21]. Tonsillectomy is one of the most common treatment methods [22].

The aim of the study was to determine the prevalence of HPV infection in patients with HNSCC, patients with chronic tonsillitis, and healthy individuals, and to establish high- or low-risk HPV genotypes in these groups. The objectives also comprised the delineation of the relationship between high- and low-risk HPV infection and clinicopathological and demographic characteristics of the study groups.

## 2. Materials and Methods

### 2.1. Study Population and Specimen Collection

This study was composed of 315 patients, of whom 76 subjects were diagnosed with HNSCC, 71 patients presented with chronic tonsillitis, and 168 cases were not affected with these conditions (the control group). The cancer samples were obtained from surgical resection and were histologically diagnosed as squamous cell carcinoma. Samples of tonsillar tissue from the chronic tonsillitis group were collected from patients during tonsillectomy and histologically verified. Samples of tonsillar tissue were verified as chronic tonsillitis. No dysplastic changes were found in the analyzed samples in the chronic tonsillitis group. Demographic data of the study groups are shown in Table 1.

All patients were diagnosed, and samples were collected at the Department of Otorhinolaryngology and Oncological Laryngology in Zabrze, Medical University of Silesia in Katowice. Fresh specimens were flash frozen in liquid nitrogen and stored at −80 °C prior to DNA extraction. Written informed consent was obtained from all patients. The study was conducted with the approval of the Bioethics Committee (Institutional Review Board on Medical Ethics, No. KNW/0022/KB1/49/16 and No. KNW/0022/KB1/49/II/16/17). DNA isolation, HPV detection, and type determination were conducted at the Department of Medical and Molecular Biology, Faculty of Medical Sciences in Zabrze, Medical University of Silesia. The main inclusion criteria for the HNSCC group included a diagnosis of primary squamous cell carcinoma of the oropharynx, no radio- or chemotherapy. Written informed consent to participate in the study and the age above 18 were the inclusion criteria for all groups.

### 2.2. HNSCC Group

The HNSCC group included 55 (72.4%) males and 21 (27.6%) females (mean age 57.8 ± 10.0 years). Most patients (72/76, 94.7%) were aged 40 years or more (Table 1). Fifty-three (69.7%) patients were smokers, 48 (63.2%) patients consumed alcohol and 40 (52.6%) subjects were smokers and alcohol users, while 61 (80.3%) patients either smoked or consumed alcohol. Tumor samples were assessed in accordance with the TNM-classification. The tumor stage was categorized according to the International Union Against Cancer (UICC) classification of head and neck tumors (7th Edition). Eleven patients were staged T1 (14.5%), 20—T2 (26.3%), 18—T3 (23.7%) and 27—T4 (35.5%). Lymph-node metastases were found in 35 patients, i.e., N1 in 21 (27.6%) and N2 in 14 (18.4%) subjects. Almost 70% of patients (52/76; 68.4%) were graded G2, 11 (14.5%) subjects—G1, and 13 (17.1%) subjects—G3. Of 76 tumor cases, 28 (36.8%) were related to the tongue, 17 (22.4%)—the mandible, 11 (14.5%)—the left palatine tonsil, 13 (17.1%)—the floor of the mouth, 6 (7.9%)—the soft palate, 5 (6.6%)—the right palatine tonsil, 5 (6.6%)—the cheek, and 2 (2.6%)—the retromolar trigone. In 9 patients, tumors were located in more than one region. None of the patients had received preoperative radio or chemotherapy.

### 2.3. Chronic Tonsillitis Group

The group consisted of 71 patients (mean age 33.4 ± 11.1 years), including 34 (47.9%) men and 37 (52.1%) women. Consumption of tobacco and alcohol was related to 10 (14.1%) and 21 (29.6%) patients, respectively. 6 (8.5%) subjects were both smokers and alcohol users and 25 (35.2%) either smoked or drank alcohol (see Table 1).

### 2.4. Control Group

Buccal epithelial scrapings were collected from the control group (mean age 39.7 ± 16.1 years), i.e., 112 women (66.7%) and 56 men (33.3%). 121 (72.0%) individuals were alcohol users and 31 (18.5%) were smokers, 29 (17.3%) were both smokers and alcohol users, while 123 (73.2%) individuals either smoked or consumed alcohol (see Table 1). None of the individuals from the control group had a history of any cancer disease or chronic tonsillitis.

### 2.5. DNA Preparation

DNA was extracted from the tumor specimens and tonsillar tissue samples using a Gene Matrix Tissue DNA Purification Kit (EURx, Gdansk, Poland) according to the manufacturer’s instructions. Before extraction, samples were slowly thawed and homogenized in FastPrep^®^-24 (MP Biomedicals, Santa Ana, CA, USA) with ceramic beads (Lysing Matrix A, MP Biomedicals, Irvine, CA, USA). DNA was extracted from buccal epithelial cells of the control group using a GeneMATRIX Swab-Extract DNA Purification Kit (EURx, Gdańsk, Poland) using standard procedures. Qualitative and quantitative analysis of the isolated DNA was performed using a spectrophotometer (NanoPhotometer Pearl, Implen, München, Germany).

### 2.6. HPV Detection and Type Determination

HPV detection and identification of subtypes were performed by GenoFlow HPV (Human Papillomavirus) Array Test Kits (DiagCor Bioscience Inc., Hong Kong) which use biotin-labeled primers, specific probes, PCR and “flow-through” hybridization technology, which allows detection of 33 common HPV subtypes. The following are considered high-risk (16, 18, 31, 33, 35, 39, 45, 51, 52, 53, 56, 58, 59, 66, 68, 73, 82) and low-risk (6, 11, 26, 40, 42, 43, 44, 54, 55, 57, 61, 70, 71, 72, 81, 84) subtypes based on their phylogenetic and epidemiological criteria and the biological niche. The extracted DNA was mixed with PCR reagent mix and DNA Taq Polymerase provided with the kit and amplified using a Mastercycler Personal Thermal Cycler (Eppendorf, Hamburg, Germany) according to the manufacturer’s protocol. Next, the amplicons were genotyped using the “flow-through” process involving hybridization, enzyme conjugation, and colorimetric detection. Amplification controls, positive and negative controls, and hybridization controls were performed in all runs. Colored dots on the membrane indicated positive reaction and were recorded by scanning the membrane and image analysis (CapturePro Image, CaptureREAD 3.1.; DiagCor Bioscience Inc., Hong Kong). An example of the results is shown in Figure 1.

### 2.7. Statistical Analysis

The distribution of variables was evaluated by the Shapiro–Wilk test and the quantile-quantile plot. The interval data were expressed as a mean value ± standard deviation (M ± SD) in the case of normal distribution. To assess the relationship between qualitative variables, the χ^2^ or Fisher exact probability test was used. To measure the association between an exposure and an outcome occurring in the absence of the exposure, the odds ratio (OR) with 95% confidence interval (CI) was calculated. Statistical significance was set at a p value below 0.05 and all tests were two tailed. Statistical analysis was performed using Statistica 13.3 [23].

## 3. Results

### 3.1. HPV Prevalence and Its Genotypes in All Study Groups

More than 35% of all cases were positive for HPV. There were statistically significant differences in the percentage of HPV-positive cases between the chronic tonsillitis and HNSCC groups (*p* < 0.05, more in the HNSCC group: 42.1% vs. 25.4%). HPV 16, HPV 43/44 and HPV 18 were detected in all groups with HPV 16 being the most common in all groups and no significant differences were found between the three groups. HPV 43/44 was the second most prevalent type and was significantly more common in the HNSCC group than in the control group (z = 2.674, *p* < 0.001) and in the HNSCC group compared with the chronic tonsillitis group (z = 2.283, *p* < 0.05) (see Table 2 and Table 3). Considering only the chronic tonsillitis and HNSCC groups, there was a relationship between the incidence of HPV in HNSCC group and chronic tonsillitis (χ^2^ = 4.59, *p* < 0.05). The HNSCC group had a greater probability of HPV-positive status than the chronic tonsillitis group (OR = 2.1, 95%CI: 1.1–4.3).

In HNSCC cases, HPV was present in 42.1% of patients. The most common infection in the HNSCC group was HPV 16 (23 patients). HPV 18 and HPV 43/44 were identified in one (1.3%) and eight (10.5%) patients, respectively. Coinfection with HPV 16 and HPV 43/44 or HPV 73 occurred in one case each (2.6% of all cases). The prevalence of HPV in patients with chronic tonsillitis was 25.4%; HPV 16 was the most prevalent type and occurred in 22.5% of patients. Infection with HPV 43/44, HPV 18 and HPV 56 was noticed in one case, respectively. Coinfection with HPV 16 and HPV 56 was noted in 1.4% of subjects. In the control group, the prevalence of HPV was 37.5%, and the most common types were HPV 16 (32.7%), followed by both HPV 43/44 and HPV 18 (2.4%), HPV 35 (1.2%), and one case (0.6%) of HPV 51 and HPV 73. 2.4% of HPV-positive samples had been infected with two types (HPV 16 and HPV 43/44, or HPV 35, or HPV 51 or HPV 18).

### 3.2. HPV Prevalence and Its Genotypes, Demographic Parameters, and Smoking and Drinking in All Study Groups

A summary of the number of HPV-positive patients depending on age, sex and addictions in the HNSCC group is given in Figure 2.

In the HNSCC group, no relationship was found between HPV-positive status and age ≥ 40 years (χ^2^ = 0.51, *p* = 0.476), sex (χ^2^ = 0.19, *p* = 0.662), smoking (χ^2^ = 2.81, *p* = 0.094), alcohol consumption (χ^2^ = 0.34, *p* = 0.560), or the use of both stimulants (χ^2^ = 0.15, *p* = 0.695). On the other hand, a relationship was found between the use of a single stimulant (either tobacco or alcohol) (χ^2^ = 4.63, *p* < 0.05). In the group where stimulants were not used, the probability of a positive HPV status was higher than in the group of smokers or alcohol users (OR = 3.54, 95%CI: 1.1–11.7). Moreover, smokers showed a decreased HPV 16 infection rate (*p* < 0.05, OR = 3.1, 95%CI: 1.1–8.9). A detailed summary of the relationships of qualitative variables in individual study groups is presented in Table 4.

In the chronic tonsillitis and control groups, no relationship was found between HPV-positive status and sex (*p* = 0.152), (*p* = 0.091), smoking (*p* = 0.978), (*p* = 0.878), drinking (*p* = 0.846), (*p* = 0.894), smoking and drinking (*p* = 0.983), (*p* = 0.635), smoking or drinking (*p* = 0.705), (*p* = 0.686), respectively (see Table 4).

For total HPV status and HPV 16 and HPV 43/44 variants, the age of the study participants considering positive and negative results was analyzed (Table 5). Statistically significant differences were found only for HPV 16 in the control group (*p* < 0.05), with older subjects being HPV-negative. In the remaining groups, differences in age between positive and negative participants were not reported. Due to the limited occurrences of HPV 18, 35, 51, 56 and 73 in the study, the results for those variants were not included.

No difference was found between the percentage of alcohol users in the HNSCC group and controls. However, the difference was observed between the HNSCC and chronic tonsillitis groups (χ^2^ = 17.05, *p* < 0.001) and between the chronic tonsillitis and control groups (χ^2^ = 14.82, *p* < 0.001). When the chronic tonsillitis and HNSCC groups were compared, alcohol users were predominant in the HNSCC group (OR = 20.3, 95%CI: 4.2–98.2) and in the control group (OR = 8.0, 95%CI: 2.6–25.1). The use of both stimulants promoted a higher prevalence of the disease in the HNSCC group compared to the control group (χ^2^ = 9.89, *p* < 0.01, OR = 4.3, 95%CI: 1.7–10.8).

An additional analysis considering drinking and smoking showed higher HPV incidence rates in the HNSCC group compared to the chronic tonsillitis group (χ^2^ = 9.33, *p* < 0.01, OR = 5.7, 95%CI: 1.8–17.9) and in the control group compared to the chronic tonsillitis group (χ^2^ = 13.9, *p* < 0.001, OR = 6.4, 95%CI: 2.3–18.0).

Additionally, the percentages of HPV-positive participants (considering HPV variants) were compared for all groups and no differences were found between sexes in any of the groups or for any HPV type (Table 6). Moreover, for all groups, HPV 43/44 was more frequently observed in smokers (*p* < 0.05, OR = 0.17, 95%CI: 0.1–0.6).

### 3.3. Associations of HPV Status and Its Genotype, Tumor Localization, TNM and Grading in HNSCC Cases

Considering tumor localization in HNSCC cases, HPV was located in 42.9% (12/28) cases in the tongue, 23.5% (4/17) in the mandible, 63.6% (7/11) in the left palatine tonsil, 38.5% (5/13) in the floor of the mouth, 33.3% (2/6) in the soft palate, 60.0% (3/5) in the right palatine tonsil, and in 20.0% (1/5) in the cheek. Other localizations were HPV negative. The results showed a statistically significant association between HPV 16 and the T stage. HPV 16 positive patients had lower odds of a T4 tumor (*p* < 0.05, OR = 3.6, 95%CI: 1.1–12.2), lower odds of tumors in the mandible (*p* < 0.05, OR = 9.5, 95%CI: 1.2–76.8) and higher odds of tumors in the palatine tonsil (left and right combined, *p* < 0.05, OR = 4.2, 95%CI: 1.3–13.4). The association between HPV 43/44 and T4 stage (*p* < 0.05) showed that the odds of a T4 tumor were greater in patients with HPV 43/44 in tumor tissue (OR = 6.7, 95%CI: 1.3–36.1). No association was found for lymph node status (N) or grading for HPV 16 or HPV 43/44. Due to the limited prevalence of HPV 18 and 73 in the study, the results for those variants were not included.

## 4. Discussion

The history of research on HPV variants suggests that although high- and low-risk HPVs are genetically related, their pathogenic nature is diverse. The variable frequency of HPV types in tumors shows the different propensity for viral gene expression to become deregulated with distinct oncogenic capacities [24,25]. In many individuals, immune activity leads to clearance of the virus, or the virus is in a latent or asymptomatic state, but some fail to resolve HPV infection and the viral genome persistence and deregulated gene expression underlie HPV-associated pathology. Of over 200 prevalent HPV types, low-risk types cause only benign and transient lesions or asymptomatic infections, and, in the general population, cancer in association with low-risk types is uncommon and requires host immune susceptibility to allow elevation of viral gene expression and malignant potential [25]. The progression routes of oncogenic and non-oncogenic HPV types towards cancer are different. Although not considered significant causative factors of malignant transformation, mainly low-risk types of HPV 6 and 11 are detected in cervical and HNSCC cancers [12,26,27,28]. Interestingly, studies on oral squamous cell carcinoma (OSCC) demonstrated the oncogenic potential of low-risk HPV 70 increased by the expression of HPV-E7 protein and inactivation of pRb [26].

Detection of HPV is performed using different methods, including signal amplification assays, assays based on hybridization, real-time PCR, microarrays, and next-generation sequencing [29,30]. The methodology used in our study detects 33 subtypes, including the non-oncogenic HPV 43/44, while other assays detect only oncogenic genotypes and fewer non-oncogenic genotypes, without HPV 43/44 [31,32,33,34,35,36,37,38,39]. Some researchers have used tests to detect all known high-risk and low-risk genotypes. However, HPV 43/44 was not observed [37,40,41,42,43,44,45,46,47,48,49]. There are no guidelines for HPV testing as regards head and neck cancer, and the methods to detect HPV DNA in oral samples have not been standardized yet [37].

### 4.1. HPV Infection in HNSCC

Our results support the epidemiological evidence that some head and neck cancers are caused by HPV and that the main risk factors of HNSCC are tobacco and alcohol abuse [1]. The overall prevalence of HPV in the HNSCC group was 42.1%, which is in line with a large number of reports of HPV DNA (2.4–74%) in head and neck cancer patients [8,31,32,33,40,50,51,52,53,54,55,56,57,58,59]. The data concerning HPV infection in HNSCC patients in Poland are very inconclusive. One study showed that HPV was not found [1] but others reported 14% HPV-positive samples [34], 20.65% from South-Central Poland [35], and 29.2% from the South-Eastern region [43]. HPV 16 was the most prevalent genotype in the present study, which is consistent with other reports [8,33,34,35,40,43,51,52]. Of note, the carcinogenic potential of HPV 16 has been described as variable and depending on the virus variant [1]. Other high-risk genotypes in our study were HPV 18 and 73. Interestingly, the second most prevalent genotype was low-risk HPV 43/44, whereas in other studies it was HPV 18 [8,34]. Database searching (PubMed and Medline) revealed no other reports showing these types in HNSCC, except one study where LR-HPV 44 in association with HR-HPV 31 was detected in buccal cancer outside the Waldeyer’s ring area [26]. Other studies reported HPV 44 in different grades of cervical lesions, in low-grade cervical intraepithelial neoplasia [60], in atypical squamous cells of undetermined significance [61], and in low-grade squamous intraepithelial lesions [62] and occasionally in unusual anogenital malignancies [63], and HPV 43 in aggressive skin cancer, sebaceous carcinomas, which are biologically similar to HNSCC [64]. Generally, HPV 43 has been isolated from mucous or cutaneous tissues and lesions and its potential role in malignant lesions is less established. HPV 44 belongs to a mucosotropic group of HPVs found in benign and malignant lesions of the anogenital tract and fortuitously in tissues and lesions of the oral cavity, oropharynx, and esophagus [27]. A number of samples in our study contained more than one HPV type, which is in line with some [32,35,42,43,44,57,65] but not all studies [33].

Our results show that the contribution of HPV to head and neck cancer is substantial, but heterogeneous with respect to smoking or drinking habits, cancer site, and T stage. Smoking and alcohol abuse increase the risk for HNSCC [66]. Typically, HPV-positive HNSCC patients are not smokers or alcohol users, but smoking increases the risk of infection with HPV [67] and chemical as well as viral factors may be involved in HNSCC carcinogenesis [12]. Patients with HPV-associated tumors who have never smoked have better clinical outcomes following standard therapies than patients with tobacco-related carcinomas [68,69]. In the HNSCC group, the probability of positive HPV status was higher in non-smoking or non-drinking patients. The prevalence of HPV 16 infection was lower in smokers in the HNSCC group, but of all participants, smokers showed an increased rate of infection with HPV 43/44. The results of other studies regarding this connection are diverse. Tumors with active HPV infection are equally frequent in smokers and non-smokers, while HPV-negative tumors are common in heavy smokers [35]. HPV DNA loads were higher in non-smoking patients with HNSCC compared to smoking subjects with HNSCC [41]. Smoking was associated with HPV-DNA positivity in HNSCC [55] and smokers accounted for the majority of patients infected with HPV 16 and/or Epstein–Barr virus (EBV) in oral and oropharyngeal squamous cell carcinoma in the South-Eastern Polish population [43]. In a group of HPV-positive patients with oropharyngeal cancer, the risk of death increased significantly with pack-years of tobacco smoking [70]. The risk of neoplastic transformation is increased by tobacco products, which could potentiate the effects of HPV 16 [71], and smoking significantly increased the risk of persistent oral HPV infection, which, in turn, might increase the risk of cancer [72]. Head and neck epithelium is often exposed to tobacco carcinogens and might interact with HPV to stimulate neoplastic lesions [71].

Recent studies support the idea that specific anatomical sites play a pivotal role in determining susceptibility to HPV infection. The highest rates of HPV infection are at the base of the tongue and palatine tonsils, followed by the oral cavity, larynx, and sinonasal mucosa [1,24,32,35,40,50,73,74]. In our study, HPV 16 was more frequently observed in tumors in the palatine tonsil and less frequently in the oral cavity region (mandible), which is in line with the data that HPV 16 is more related to oropharyngeal primaries than to those located in the oral cavity [75]. However, one study reported that the oral cavity was more frequently infected by HPV [42] and another study showed that HPV was slightly more prevalent in the oral cavity than in oropharyngeal cancer [43].

In many countries, some studies reported the trends which showed that HNSCC patients infected with HPV 16 were younger and that survival was better than in the absence of HPV [1,36,43,68,75]. Patients infected with uncommon subtypes (i.e., other than HPV 16) survived 5 years without cancer progression [35]. However, in some studies, HPV infection was observed more frequently in advanced stages of tumors [35,42], as confirmed in our study. However, only HPV 43/44 infection was associated with advanced T stage and the opposite trend was reported in patients with HPV 16 infection. Janecka-Widła et al. found that over 59% of HPV-positive HNSCC cases had stage T3, but there was no significant relationship between HPV infection and the T stage in oropharyngeal cases. However, over 55% of them had also stage T3 [35]. Chen et al. observed that HPV 16 was correlated with the T stage [76]. As in our results, Ziai et al. showed that patients positive for HPV 16 tended to be younger and less likely to have T4 stage [75]. Among HPV-positive patients, the G1 grade was more prevalent than in HPV-negative subjects, and G3 was more prevalent in HPV-negative patients [43]. On the other hand, in HNSCC patients, no significant relationships were found between the presence of HPV and patients’ age, nodal status, grade, tumor stage, smoking, or alcohol consumption [31,35,44,55,59,77]. Our tumor samples were collected for two years. Due to a short period of time, we were not able to make the analyses regarding overall survival and disease-free survival. In the future, we plan to perform such analyses.

Among low-risk HPV 40, 42, 43, and 44, genomic variants were identified [78]. Of note, the low-risk HPV 43/44 was observed frequently in this study. This genotype has been rarely assessed and thus has been overlooked as a possible risk factor. Moreover, we noticed that HPV 43/44 was frequently observed in patients with advanced tumor stage T4, which suggests that this variant could be involved in higher progression. Furthermore, HPV 43/44 was observed frequently in smokers. Tobacco use induces proinflammatory and immunosuppressive effects and may increase predisposition to HPV infection [79,80]. Current understanding of the steps leading from HPV infection to cancer remains mainly limited to the model of cervical cancer [2], and is complicated by different oncogenic phenotypes of different HPV genome variants [81,82]. Based on our observations, we favor the combined etiology and suggest that additional risk factor(s) such as smoking act in a synergic manner with HPV infection, even in the cases of low-risk HPV in carcinogenesis. Larger prospective studies to explore the significance of HPV 43/44 infection in HNSCC are warranted to provide a basis for clinically relevant classification of LR-HPV types, which could be considered in treatment and prognosis of patients.

### 4.2. HPV Infection in Chronic Tonsillitis

The results on HPV prevalence in tissue from chronic tonsillitis vary from 0% to 21% [29,33,41,83,84,85,86,87,88,89]. In our study, 25.4% of samples were positive for HPV, and HPV 16 was the most prevalent genotype, which is in line with other reports [33,41]. In non-cancerous tonsillar hyperplasia, 28.0% of samples were HPV DNA-positive [55], while 13.1% of patients undergoing tonsillectomy due to benign indications showed HPV-positive gargle samples [24]. Other genotypes detected in chronic tonsillitis in our study were HPV 43/44, HPV 18 and HPV 56. Several other studies reported HR and LR types of HPV in patients undergoing tonsillectomy, but HPV 43/44 was not previously detected [24,29,40,85,86,88]. Of note, oncogenic HPV types have been detected mainly in children and young adults [85,86]. In our study, some chronic tonsillitis samples showed multiple HPV types, which is in line with other studies [24,85], and although we detected HR-HPV in the study population, this part of our study suggests that the etiology of chronic tonsillitis is not related to HPV. Our intention in the future is to continue the follow-up of this group of HPV-positive patients and to conduct studies on a larger population.

### 4.3. HPV Infection in Healthy Individuals

One interesting finding of our study is the high prevalence (37.5%) of HPV infection in the normal oral mucosa of healthy individuals. Other studies of cancer-free individuals reported values from 0% to 81.1% in normal oral mucosal cells, exfoliated cells from normal tonsils, or oral gargles and tonsillar washings [37,38,45,46,86,90,91,92,93]. In Polish population, a low frequency (1.08%) was observed in normal oral squamous cells [38], while a higher prevalence in oropharyngeal swabs was observed in couples (19.8% in women and 28.4% in men) and in 5.6% of couples, both partners were infected (in 81.8% of cases with identical HPV types). That study revealed frequent infection with LR-HPV 42 [47]. HPV 16 was the most prevalent genotype, and its prevalence and the distribution of other genotypes generally reflect those reported in other studies [46,80,91], except for HPV 43/44 which we reported for the first time in oral mucosal epithelial samples in the Polish population, although it was observed earlier in cervical and anal swab samples from healthy populations [61,93,94,95,96,97,98]. Moreover, sequencing analysis of the HPV E6 region from endocervical swab samples revealed the highest nucleotide substitutions and amino acid changes for HPV 44 genotype compared with other genotypes [94]. Infections with more than one genotype comprised 2.4% of individuals, as observed in several previous studies [46,47,48,91,93].

The limitation of the study is the small size of the groups. Further studies on a larger scale are needed to assess the HPV types in the control group, using non-invasive methods such as oral swabs or salivary rinses. Moreover, further studies are warranted to characterize the route of transmission and time of persistent and latent HPV infection in normal oral mucosa.

No associations between HPV status and age, sex, smoking or drinking habits were found in the group with chronic tonsillitis and healthy individuals, which is in line with other studies [38,91]. However, the mean age of patients in the control group infected with HPV 16 was lower (Table 5). Kreimer et al. [80] showed that newly acquired oral oncogenic HPV infections in healthy men were rare and constant across age groups, although other studies reported that the incidence increased with age and/or tobacco use [45,46], which might be caused by a longer duration of infections rather than by a higher incidence [80].

The transmission of HPV is both sexual and non-sexual [90], and oral infection is significantly more prevalent in men than in women and increases with the number of lifetime sex partners [37,45]. Oral infections may play a crucial role in the transmission of HPV DNA among family members [90]. The prevalence of oral HPV in partners of patients with oropharyngeal cancer was 15%, and the authors concluded that they could spread oral HPV and should be studied prospectively to identify oral HPV persistence and estimate the risk of cancer [99]. Since the presence of HPV does not necessarily mean an active infection, the detection of HPV DNA does not imply that the infection was detected [27]. In women with oral HPV infection, 46.2% cleared their infection at different times for various HPV genotypes [48]. The reasons why some infected people experience only asymptomatic infection are not entirely understood. However, they are most likely related to reduced or lack of immune response to infection in a certain group of patients. A very high percentage of HPV infection was found in the cervix and oral cavity in HIV-positive women [100].

In our study, we detected seven HPV subtypes in total, including high-risk HPV 16, 18, 35, 51, 56, 73, and one low-risk type (HPV 43/44) found in all groups, including HNSCC patients. Discrepancies between our data and other published data [29,37,39,85,88,91] could result from different methods used to detect HPV, in the study population, especially regarding different tumor locations and smoking and/or alcohol abuse, in heterogeneity of sampling methods, and/or in geographical distribution of HPV genotypes [24,35,37,40,50]. Furthermore, they may be connected to different transmissions routes of different HPV types and the need to produce the appropriate number of copies of the virus to infect a host [101]. Although HPV is described as causing virus-associated cancers, some individuals with HPV develop cancer, while others are cleared from the infection. For instance, most human oral papillomavirus infections are asymptomatic and are naturally cleared by the immune system, at different rates for various genotypes [48,80]. Moreover, HPV types of low neoplastic potential are cleared from the oral mucosa more quickly than the high-risk types [48]. The significance of the detected quantity of HPV DNA is therefore unclear because the presence of even a small copy number cannot rule out the risk of malignant transformation [80].

### 4.4. Relevance to Vaccination Programs

There is a need for educational intervention to increase knowledge about HPV and HPV-associated cancer and transfer effective recommendations for HPV vaccination, which is a complex issue [102,103]. HPV vaccines have high efficacy against infection and disease caused by HPV and could potentially reduce the risk of HPV-dependent oral cancers [104]. It is estimated that approximately 120 million young women have been targeted by HPV vaccination programs, but only 1% of them lived in less developed countries [74]. Only 2 of 168 (1.19%) individuals from the healthy control group had been vaccinated, and HPV was not detected in vaccinated female participants. There were no data for the HNSCC group or the chronic tonsillitis group. However, we suspect that almost none of them was vaccinated. Such a low number of vaccinated participants may reflect the fact that the three different vaccines of the National Vaccination Program are not reimbursed in Poland [67]. The access to HPV testing and vaccination should be improved in many countries based on the evidence that HPV vaccination introduced in other countries a decade ago showed excellent results and decreased the prevalence of cervical and oral HPV infections [105,106,107]. Moreover, based on the modelling global strategy to accelerate cervical cancer elimination, the results showed that elimination of cervical cancer was possible by the end of the century using vaccination against HPV with a high coverage and more intense cervical screening, especially in countries with the highest risk [108]. It is important to promote more widespread HPV vaccination for prevention against HPV infection and HPV-associated cancers. Vaccination of boys is recommended only in a few countries [109]. Furthermore, to reach the goal, the efforts should also focus on promoting HPV vaccine trials among men [110].

## 5. Conclusions

The epidemiology of HPV infection is related to many complicated issues and the HPV-related carcinogenesis is still controversial. Our analysis provides insight into the prevalence of oral oncogenic and non-oncogenic HPVs in patients with HNSCC, patients with chronic tonsillitis, and healthy individuals. HPV 16 was the genotype with the highest incidence, and we found that non-oncogenic HPV 43/44 was frequent in HNSCC patients, which was not previously confirmed. An oncogenic potential for HPV 43/44 can be suggested but further and more extensive studies are warranted to confirm it. 

Our study shows that further investigations are necessary to examine larger number of patients with attention paid to high and low-risk HPV genotypes. Moreover, it seems reasonable that the presence of any HPV type in patients with HNSCC should be considered during treatment planning. There is a need for larger analyses if LR-HPV infection is a risk factor for the survival outcome. In our study, cancer specimens were obtained during a 2-year period. We plan to analyze the prognostic impact of the HPV status on the potential 3- and 5-year overall survival and 3- and 5-year disease-free survival in HNSCC patients.

The prevalence of high-risk HPV in the mucous membrane of healthy individuals is high and additional studies are warranted to estimate the progression rate from HPV infection to HPV-associated HNSCC. Despite the undisputed significance of protective HPV vaccination programs, the data from a low percentage of vaccinated people showed that such programs were not sufficient for prevention and efforts should be focused on screening and effective detection of HPV. We believe that HPV infection can be prevented through extensive and free vaccination programs, and the creation of an effective platform for HPV testing. Furthermore, our observations lead to the suggestion that HPV vaccination is of considerable importance in public health strategies. A panel of vaccines for different types of HPV, including low-risk types, should be adopted for vaccination of male and female populations.

## Figures and Tables

**Figure 1 diagnostics-11-02180-f001:**
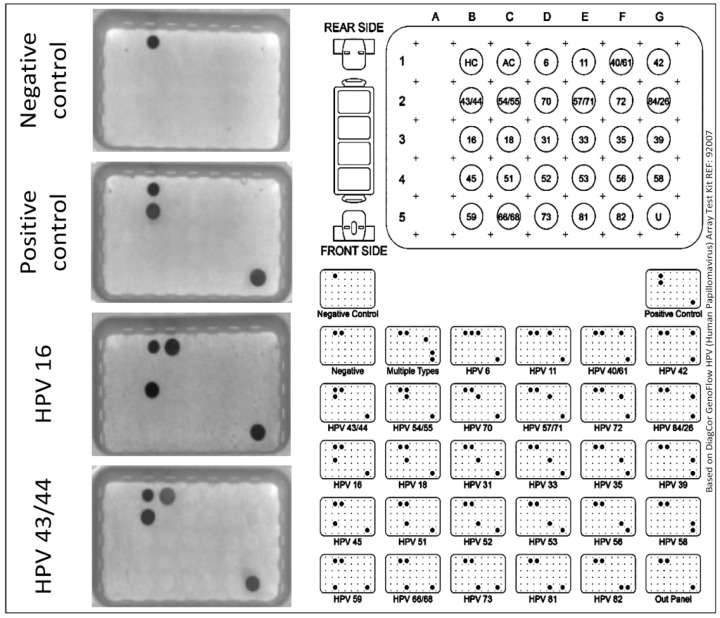
Example of HPV detection by hybridization, with the results for negative and positive controls and HPV 16 and HPV 43/44 positive samples.

**Figure 2 diagnostics-11-02180-f002:**
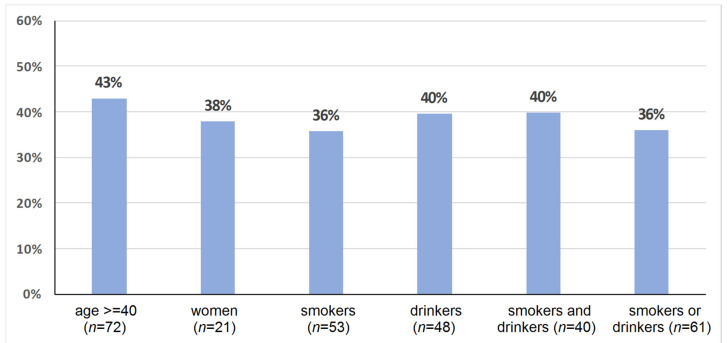
Percentage of HPV-positive results depending on qualitative variables (age, sex, addictions) in the HNSCC group (N = 76). The total number of study cases is given in parentheses next to the variable.

**Table 1 diagnostics-11-02180-t001:** Demographic data of the study groups.

	All		HNSCC		Chronic Tonsillitis		Control	
	Mean	SD	Mean	SD	Mean	SD	Mean	SD
Age	42.66	±16.39	57.83	±10.02	33.35	±11.11	39.73	±16.06
	*n*	%	*n*	%	*n*	%	*n*	%
Age ≥ 40	167	53.0	72	94.7	18	25.4	77	45.8
Age < 40	148	47.1	4	5.3	53	74.7	91	54.2
Female	170	54.0	21	27.6	37	52.1	112	66.7
Male	145	46.0	55	72.4	34	47.9	56	33.3
Smokers	94	29.8	53	69.7	10	14.1	31	18.5
Non-smokers	221	70.2	23	30.3	61	85.9	137	81.6
Alcohol users	190	60.3	48	63.2	21	29.6	121	72.0
Alcohol non-users	125	39.7	28	36.8	50	70.4	47	28.0
Smokers and alcohol users	75	23.8	40	52.6	6	8.5	29	17.3
Smokers or alcohol users	209	66.4	61	80.3	25	35.2	123	73.2

**Table 2 diagnostics-11-02180-t002:** The number of positive HPV results in all analyzed groups, considering all HPV variants. The percentages of HPV-positive results in each group are given in parentheses.

	All N = 315	HNSCC N = 76	Chronic Tonsillitis N = 71	Control N = 168
	*n* (%)	*n* (%)	*n* (%)	*n* (%)
HPV-positive	113 (35.9)	32 (42.1)	18 (25.4)	63 (37.5)
HPV 16	94 (28.8)	23 (30.3)	16 (22.5)	55 (32.7)
HPV 43/44	13 (4.1)	8 (10.5)	1 (1.4)	4 (2.4)
HPV 18	6 (1.9)	1 (1.3)	1 (1.4)	4 (2.4)
HPV 35	2 (0.6)	-	-	2 (1.2)
HPV 51	1 (0.3)	-	-	1 (0.6)
HPV 56	1 (0.3)	-	1 (1.4)	-
HPV 73	2 (0.6)	1 (1.3)	-	1 (0.6)

**Table 3 diagnostics-11-02180-t003:** Statistical significance calculated with the proportion test of the HPV-positive results between groups.

Groups	HPV-Positive	HPV 16	HPV 43/44	HPV 18	HPV 35	HPV 51	HPV 56	HPV 73
Control vs. HNSCC	0.497	0.704	<0.001	0.589	0.341	0.503	-	0.562
Chronic tonsillitis vs. HNSCC	<0.05	0.289	<0.05	0.960	-	-	0.298	0.332
Control vs. chronic tonsillitis	0.070	0.114	0.631	0.631	0.357	0.516	0.124	0.516

**Table 4 diagnostics-11-02180-t004:** Analysis of demographic variables in study groups depending on the occurrence of HPV. Statistical significance (*p*) was calculated by the χ^2^ test.

Study Group	Variable	Variant	HPV-Positive	HPV-Negative	χ^2^	*p*
HNSCC N = 76	sex	women	8	13	0.19	0.662
		men	24	31		
	smoking	yes	19	34	2.81	0.094
		no	13	10		
	drinking	yes	19	29	0.34	0.560
		no	13	15		
	smoking and drinking	yes	16	24	0.15	0.695
		no	16	20		
	smoking or drinking	yes	22	39	4.63	<0.05
		no	10	5		
Chronic tonsillitis N = 71	sex	women	12	25	2.04	0.152
		men	6	28		
	smoking	yes	3	7	0.001	0.978
		no	15	46		
	drinking	yes	5	13	0.04	0.846
		no	16	37		
	smoking and drinking	yes	1	17	0.0004	0.983
		no	5	48		
	smoking or drinking	yes	7	11	0.14	0.705
		no	18	35		
Control N = 168	sex	women	47	65	2.86	0.091
		men	16	40		
	smoking	yes	12	19	0.02	0.878
		no	51	86		
	drinking	yes	45	76	0.02	0.894
		no	18	29		
	smoking and drinking	yes	12	17	0.23	0.635
		no	51	88		
	smoking or drinking	yes	45	78	0.16	0.686
		no	18	27		

**Table 5 diagnostics-11-02180-t005:** Summary of mean values and standard deviation for age (years) in individual groups, considering the type of HPV and the results (negative, positive).

Group	Variant	HPV Status	HPV 16	HPV 43/44
HNSCC	Positive	58 ± 10	59 ± 11	56 ± 7
	Negative	58 ± 10	57 ± 10	58 ± 10
	*p*-value	0.748	0.277	0.313
Chronic tonsillitis	Positive	30 ± 9	20 ± 9	
	Negative	35 ± 12	34 ± 12	
	*p*-value	0.071	0.177	
Control	Positive	37 ± 15	36 ± 14	29 ± 2
	Negative	41 ± 17	42 ± 17	40 ± 16
	*p*-value	0.083	<0.05	0.165
All	Positive	42 ± 17	41 ± 17	45 ± 16
	Negative	43 ± 16	44 ± 16	43 ± 16
	*p*-value	0.434	0.116	0.599

**Table 6 diagnostics-11-02180-t006:** The number of HPV-positive results depending on sex in individual study groups, considering the HPV variant. The percentages of HPV-positive results are given in parentheses. Statistical significance (*p*) was calculated using the proportion test.

Study Group		HPV-Positive	HPV 16	HPV 43/44	HPV 18	HPV 35	HPV 51	HPV 56	HPV 73
HNSCC	Women	8 (38.1%)	5 (23.8%)	2 (9.5%)	1 (4.8%)	-	-	-	1 (4.8%)
	Men	24 (43.6%)	18 (32.7%)	6 (10.9%)	0 (0%)	-	-	-	0 (0%)
	*p*	0.664	0.452	0.861	0.106	-	-	-	0.106
Chronic tonsillitis	Women	12 (32.4%)	10 (27.0%)	1 (2.7%)	1 (2.7%)	-	-	0 (0%)	-
	Men	6 (17.7%)	6 (17.7%)	0 (0%)	0 (0%)	-	-	1 (2.9%)	-
	*p*	0.155	0.348	0.338	0.338	-	-	0.297	-
Control	Women	47 (42.0%)	42 (37.5%)	2 (1.8%)	3 (2.7%)	2 (1.8%)	0 (0%)	-	0 (0%)
	Men	16 (28.6%)	13 (23.2%)	2 (3.6%)	1 (1.8%)	0(0%)	1 (1.8%)	-	1 (1.8%)
	*p*	0.092	0.064	0.476	0.721	0.316	0.157	-	0.157
All	Women	67 (39.4%)	57 (33.5%)	5 (2.9%)	5 (2.9%)	2 (1.2%)	0 (0%)	0 (0%)	1 (0.6%)
	Men	46 (31.7%)	37 (25.5%)	8 (5.5%)	1 (0.7%)	0 (0%)	1 (0.7%)	1 (0.7%)	1 (0.7%)
	*p*	0.157	0.122	0.253	0.146	0.191	0.279	0.279	0.910

## Data Availability

The data used to support the findings of this research are available upon request.

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
