# Peer review of "The Prevalence of High- and Low-Risk Types of HPV in Patients with Squamous Cell Carcinoma of the Head and Neck, Patients with Chronic Tonsillitis, and Healthy Individuals Living in Poland"

_diagnostics, 2021, doi:10.3390/diagnostics11122180_

Round 1
Reviewer 1 Report
In the presented work, the authors investigate the prevalence of HPV infection in patients with head and neck squamous cell carcinoma (HNSCC), patients with chronic tonsillitis, and healthy individuals, and to establish the HPV genospecies with high- and low-risk in these groups. This theme is widely discussed in the world literature.
The structural elements of the text of a scientific article are logically interconnected. It is necessary to note the high level of elaboration of the research problem.
The section "Material and methods" indicates the number of patients in the study sample, n = 315, which is sufficient to obtain correct results. The presence of a control group in the presented study is undoubtedly a valuable criterion. The clinical and morphological parameters of thepatients included in the study are described in detail, which is an positive point. DNA preparation, HPV detection and type determination, Statistical methods are described below, which gives reason to believe that the results can be fully reproduced. It is important to note that a fairly large range of HPV genotypes is being investigated, which is also an advantage of the presented work.
However, for patients with chronic tonsillitis, it is necessary to indicate whether the presence of dysplastic changes has been confirmed or refuted.
The section "Results" contains detailed descriptions of the results of the study, the processing of which is beyond doubt.
Separately, it is necessary to note the presentation of the "Discussion" section, which describes the significance of the results obtained and the literature data for each separately taken study group, as well as "Relevance to vaccination programs".
However, it would be interesting for the HNSCC group to add an assessment of the outcome of the disease depending on the HPV status. That is, in this group of patients, it would be interesting to discuss the relationship between HPV status and the effect of chemotherapy or radiation therapy, if patients received treatment.
Also, HPV + samples need confirmation by another method. For example, you can use immunohistochemical methods to determine the presence of HPV.
And also to expand the literature data on vaccination results in the section "Relevance to vaccination programs".
When analyzing the literary sources, it was shown that only 45% of the sources were for the last 5 years. Perhaps the authors should use more relevant literature data, given the fairly large amount of information in this area of research.
Author Response
Dear Reviewers,
I highly appreciate your effort related to the improvement of the manuscript.
I am grateful for all the suggestions of the Reviewers’.
The responses to the Reviewers’ comments are attached below, and the corresponding changes are shown in red in the revised manuscript.
RESPONSE TO REVIEWER 1
Thank you very much for reviewing the manuscript. We truly appreciate all suggestions and comments that helped us improve our work.
- „However, for patients with chronic tonsillitis, it is necessary to indicate whether the presence of dysplastic changes has been confirmed or refuted”
Thank you for your highly professional comments. We added this information to the manuscript: ”No dysplastic changes were found in the analyzed samples in the chronic tonsillitis group”.
- „However, it would be interesting for the HNSCC group to add an assessment of the outcome of the disease depending on the HPV status. That is, in this group of patients, it would be interesting to discuss the relationship between HPV status and the effect of chemotherapy or radiation therapy, if patients received treatment”
Thank you for your highly professional comments. None of the patients had received preoperative radio- or chemotherapy. In our study, cancer specimens were obtained during a 2-year period. Due to the lack of data on survival outcomes, we could not perform further analysis on this issue. We plan to analyze the prognostic potential for 3- and 5-year overall survival (OS) and 3- and 5-year disease-free survival (DFS). It should be emphasized that 2-3-year survival provides less information than 5-year survival, which is standard in oncology and a source of reliable knowledge of survival outcome data. In the future, we plan to check if the presence of HPV defines the response to chemotherapy or radiotherapy.
We added this information to our manuscript.
- Also, HPV + samples need confirmation by another method. For example, you can use immunohistochemical methods to determine the presence of HPV.
HPV detection and identification of subtypes were performed by GenoFlow HPV (Human Papillomavirus) Array Test Kits (DiagCor Bioscience Inc., Hong Kong) which use biotin-labeled primers, specific probes, PCR and "flow-through" hybridization technology, which allows detection of 33 common HPV subtypes. GenoFlow HPV (Human Papillomavirus) Array Test Kits (DiagCor Bioscience Inc., Hong Kong) is recommended for diagnostic use.
We also used the scientific control as recommended by attached instructions of test in each run. The scientific control was designed to monitor the test’s performance. In each run, we used amplification controls, positive and negative controls, and also hybridization controls.
Moreover, the GenoFlow HPV Array Test Kit has Universal Signal (U), the sample is double checked in each run.
Additionally, for our internal control in our study, we used also HPV 16/18-FRT PCR kit (AmpliSens), REF: R-V12(RG,iQ,Mx)-CE; LightCycler 480 (Roche) for a few selected samples for a more thorough and broader study. Our results were confirmed by the second test. The data are collected in our database and could be shown upon request.
The purpose of the study was not to compare methods of HPV detection. After searching the literature, we chose the test with high sensitivity and specificity and the test that was less labor-intensive.
- Wong OG, Lo CK, Chow JN, Tsun OK, Szeto E, Liu SS, Ngan HY, Cheung AN. Comparison of the GenoFlow human papillomavirus (HPV) test and the Linear Array assay for HPV screening in an Asian population. J Clin Microbiol. 2012 May;50(5):1691-7. doi: 10.1128/JCM.05933-11. Epub 2012 Feb 15. PMID: 22337983; PMCID: PMC3347143).
- Alhamlan FS, Khayat HH, Obeid DA, Tulba AM, Baduwais TS, Alfageeh MB, Al-Ahdal MN. Clinical comparison of two human papillomavirus detection assays: GenoFlow and reverse line blot. J Infect Dev Ctries. 2020 Jan 31;14(1):97-103. doi: 10.3855/jidc.11769. PMID: 32088690.
- Sohrabi A, Rahnamaye-Farzami M, Mirab-Samiee S, Mahdavi S, Babaei M. Comparison of In-House Multiplex Real Time PCR, Diagcor GenoFlow HPV Array Test and INNO-LiPA HPV Genotyping Extra Assays with LCD- Array Kit for Human Papillomavirus Genotyping in Cervical Liquid Based Cytology Specimens and Genital Lesions in Tehran, Iran. Clin Lab. 2016;62(4):615-9. doi: 10.7754/clin.lab.2015.150808. PMID: 27215080.
Immunohistochemical methods show less sensitivity and specificity. Moreover, fewer tests allow to detect a high number of HR and LR genotypes. Our test allowed to detect 33 common HPV subtypes.
4.And also to expand the literature data on vaccination results in the section "Relevance to vaccination programs".
Thank you for your thoughtful comments. We expanded this issue in manuscript. We added 5 references to this section:
- Walker KK, Jackson RD, Sommariva S, Neelamegam M, Desch J. USA dental health providers' role in HPV vaccine communication and HPV-OPC protection: a systematic review. Hum Vaccin Immunother. 2019;15(7-8):1863-1869. doi: 10.1080/21645515.2018.1558690. Epub 2019 Jan 30. PMID: 30620632; PMCID: PMC6746472.
- Grandahl M, Nevéus T. Barriers towards HPV Vaccinations for Boys and Young Men: A Narrative Review. Viruses. 2021 Aug 19;13(8):1644. doi: 10.3390/v13081644. PMID: 34452508; PMCID: PMC8402923.
- Fiorito TM, Krilov LR, Nonaillada J. Human Papillomavirus Knowledge and Communication Skills: A Role-Play Activity for Providers. MedEdPORTAL. 2021 Apr 23;17:11150. doi: 10.15766/mep_2374-8265.11150. PMID: 33907710; PMCID: PMC8063629.
- Enerly E, Flingtorp R, Christiansen IK, Campbell S, Hansen M, Myklebust TÅ, Weiderpass E, Nygård M. An observational study comparing HPV prevalence and type distribution between HPV-vaccinated and -unvaccinated girls after introduction of school-based HPV vaccination in Norway. PLoS One. 2019 Oct 10;14(10):e0223612. doi: 10.1371/journal.pone.0223612. Erratum in: PLoS One. 2019 Dec 12;14(12):e0226706. PMID: 31600341; PMCID: PMC6786612.
- Brisson M, Kim JJ, Canfell K, Drolet M, Gingras G, Burger EA, Martin D, Simms KT, Bénard É, Boily MC, Sy S, Regan C, Keane A, Caruana M, Nguyen DTN, Smith MA, Laprise JF, Jit M, Alary M, Bray F, Fidarova E, Elsheikh F, Bloem PJN, Broutet N, Hutubessy R. Impact of HPV vaccination and cervical screening on cervical cancer elimination: a comparative modelling analysis in 78 low-income and lower-middle-income countries. Lancet. 2020 Feb 22;395(10224):575-590. doi: 10.1016/S0140-6736(20)30068-4. Epub 2020 Jan 30. PMID: 32007141; PMCID: PMC7043009.
- „When analyzing the literary sources, it was shown that only 45% of the sources were for the last 5 years. Perhaps the authors should use more relevant literature data, given the fairly large amount of information in this area of research.”
Most studies that made a significant contribution to the knowledge about HPV distribution in patients with head and neck squamous cell carcinoma (HNSCC), patients with chronic tonsillitis, and healthy individuals were cited. The latest paper only refers to the prevalence of HPV in different countries
(https://pubmed.ncbi.nlm.nih.gov/27930766/ https://bmccancer.biomedcentral.com/articles/10.1186/s12885-021-08213-9 https://www.mdpi.com/2072-6694/11/6/820/htm https://pubmed.ncbi.nlm.nih.gov/30134232/ https://pubmed.ncbi.nlm.nih.gov/28948421/ https://pubmed.ncbi.nlm.nih.gov/30839148/ https://www.sciencedirect.com/science/article/abs/pii/S156713481830042X https://bmccancer.biomedcentral.com/articles/10.1186/s12885-017-3789-0)
Due to the fact that in our paper there is already a large number of citations regarding the frequency (14 regarding HNSCC), we focused on the topic concerning low-risk genotypes and after searching the database, we tried to add new citations not only for high-risk HPV genotypes but also for low-risk ones. We added 5 new references to the “Relevance to vaccination programs” section, which shed a new light on this issue and improved this field of science.
Reviewer 2 Report
The paper does not add interesting data for international community. Maybe for Poland these data are interesting.
One big issue is that authors used a screening assay used in cervical cancer, and did not confirm their results with any other validated method. There are almost no report of using this assay in HNSCC. This might be the reason why some data about HPV16 in healthy individuals are aberrant.
Organization of results and discussion is really heavy and difficult to read, though english is quite correct.
I think that further work to improve the paper is really really important, and it might require a lot of time.
Author Response
Dear Reviewers,
I highly appreciate your effort related to the improvement of the manuscript.
I am grateful for all the suggestions of the Reviewers’.
The responses to the Reviewers’ comments are attached below, and the corresponding changes are shown in red in the revised manuscript.
RESPONSE TO REVIEWER 2
Thank you very much for reviewing the manuscript. We appreciate all suggestions and comments that helped us improve our work.
The paper does not add interesting data for international community. Maybe for Poland these data are interesting.
We respectfully disagree with this statement. Although we performed analyses on the Polish population, HNSCC and chronic tonsillitis are common diseases in many populations worldwide. Similarly, human papillomavirus (HPV) is a virus with the potential to infect humans all over the world. The aim of the present study was not only to determine the prevalence of HPV infection in patients with head and neck squamous cell carcinoma (HNSCC), patients with chronic tonsillitis, or healthy individuals, but also establish the high- and low-risk HPV genotypes in these groups.
Our conclusion is as follows: “The epidemiology of HPV infection is related to many complicated issues and the HPV-related carcinogenesis is still controversial. Our analysis provides insight into the prevalence of oral oncogenic and non-oncogenic HPVs in patients with HNSCC, patients with chronic tonsillitis, and healthy individuals. HPV 16 was the genotype with the highest incidence, and we found that non-oncogenic HPV 43/44 was frequent in HNSCC patients, which was not previously confirmed.An oncogenic potential for HPV 43/44 can be suggested but further and more extensive studies are warranted to confirm it”.
One big issue is that authors used a screening assay used in cervical cancer, and did not confirm their results with any other validated method. There are almost no report of using this assay in HNSCC. This might be the reason why some data about HPV16 in healthy individuals are aberrant.
Thank you for your thoughtful comments. HPV detection and identification of subtypes were performed by GenoFlow HPV (Human Papillomavirus) Array Test Kits (DiagCor Bioscience Inc., Hong Kong) which use biotin-labeled primers, specific probes, PCR and "flow-through" hybridization technology which allows detection of 33 common HPV subtypes. GenoFlow HPV (Human Papillomavirus) Array Test Kits (DiagCor Bioscience Inc., Hong Kong) is recommended for the diagnostic use. In fact, the test was used for cervical cancer. From the very beginning of the research, this issue was consulted with the company (DiagCor Bioscience Inc., Hong Kong). The test is based on previously isolated DNA, and the source from which the material was obtained was of no great importance in this type of research. We worked on the isolated DNA – it is the test for the detection of HPV DNA.
Moreover, the GenoFlow HPV Array Test Kit has Universal Signal (U), the sample is double checked in each run. We also used the scientific control as recommended by attached instructions of test in each run. The scientific control was designed to monitor the test’s performance. In each run, we used amplification controls, positive and negative controls, and also hybridization controls.
Additionally, for our internal control in our study, we used also HPV 16/18-FRT PCR kit (AmpliSens), REF: R-V12(RG,iQ,Mx)-CE; LightCycler 480 (Roche) for a few selected samples for a more thorough and broader study. Our results were confirmed by the second test. The data are collected in our database and could be shown upon request.
The purpose of the study was not to compare methods of HPV detection. After searching the literature, we chose the test with high sensitivity and specificity and the test that was less labor-intensive.
- Wong OG, Lo CK, Chow JN, Tsun OK, Szeto E, Liu SS, Ngan HY, Cheung AN. Comparison of the GenoFlow human papillomavirus (HPV) test and the Linear Array assay for HPV screening in an Asian population. J Clin Microbiol. 2012 May;50(5):1691-7. doi: 10.1128/JCM.05933-11. Epub 2012 Feb 15. PMID: 22337983; PMCID: PMC3347143).
- Alhamlan FS, Khayat HH, Obeid DA, Tulba AM, Baduwais TS, Alfageeh MB, Al-Ahdal MN. Clinical comparison of two human papillomavirus detection assays: GenoFlow and reverse line blot. J Infect Dev Ctries. 2020 Jan 31;14(1):97-103. doi: 10.3855/jidc.11769. PMID: 32088690.
- Sohrabi A, Rahnamaye-Farzami M, Mirab-Samiee S, Mahdavi S, Babaei M. Comparison of In-House Multiplex Real Time PCR, Diagcor GenoFlow HPV Array Test and INNO-LiPA HPV Genotyping Extra Assays with LCD- Array Kit for Human Papillomavirus Genotyping in Cervical Liquid Based Cytology Specimens and Genital Lesions in Tehran, Iran. Clin Lab. 2016;62(4):615-9. doi: 10.7754/clin.lab.2015.150808. PMID: 27215080.
Generally, there are a lot of sources of variable results - the genotypic distribution of HPV among different population, especially regarding the selection of patients with different tumor location, and smoking and/or alcohol abuse. The causes of such variability are related to the geographical distribution of HPV genotypes. The final cause of such variability is related to sampling methods because of source heterogeneity such as oral gargles, tonsillar washings, tonsil brushings, frozen tissues, paraffin embedded tissue, and plasma.
Organization of results and discussion is really heavy and difficult to read, though english is quite correct.
Thank you for your thoughtful comments. English was professionally corrected by the assistant professor from the Department of English Studies, who is a medical translator and interpreter (we attached certificate).We added same missing information hoping that the text has become more readable.